# Dataset: Variable Message Signal Annotated Images for Object Detection

Enrique Puertas [1,*] , Gonzalo De-Las-Heras [2] , Javier Sánchez-Soriano [3] and Javier Fernández-Andrés [4]

1 Department of Science, Computing and Technology, Universidad Europea de Madrid, Calle Tajo s/n, Villaviciosa de Odón, 28670 Madrid, Spain

2 SICE Canada Inc., Toronto, ON M4P 1G8, Canada; gdelasheras@sice.com

3 Higher Polytechnic School, Universidad Francisco de Vitoria, 28223 Pozuelo de Alarcón, Spain; javier.sanchez@ufv.es

4 Department of Engineering, Universidad Europea de Madrid, Calle Tajo s/n, Villaviciosa de Odón, 28670 Madrid, Spain; javier.fernandez@universidadeuropea.es

* Correspondence: enrique.puertas@universidadeuropea.es

**Abstract:** This publication presents a dataset consisting of Spanish road images taken from inside a vehicle, as well as annotations in XML files in PASCAL VOC format that indicate the location of Variable Message Signals within them. Additionally, a CSV file is attached with information regarding the geographic position, the folder where the image is located and the text in Spanish. This can be used to train supervised learning computer vision algorithms such as convolutional neural networks. Throughout this work, the process followed to obtain the dataset, image acquisition and labeling and its specifications are detailed. The dataset constitutes 1216 instances, 888 positives and 328 negatives, in 1152 jpg images with a resolution of 1280 × 720 pixels. These are divided into 756 real images and 756 images created from the data-augmentation technique. The purpose of this dataset is to help in road computer vision research since there is not one specifically for VMSs.

**Dataset:** https://zenodo.org/record/5904211.

**Dataset License:** Creative Commons Attribution 4.0 International.

**Keywords:** variable message signal (VMS); dataset; machine learning; ADAS; PASCAL VOC; autonomous driving; deep learning; neural networks; retinanet

## 1. Introduction

Variable Message Signs (VMS) are devices used to communicate text messages and pictograms to drivers [1]. They are made up of a series of LED arrays on a black background. This type of signaling has been shown to be useful for safety as it provokes speed reduction [2,3] and helps to resolve traffic jams [4].

Several studies indicate that VMS has a positive effect on driving by reducing speed and reducing traffic congestion caused by accidents or other events. However, reading them is a distraction, which is a cause of accidents [5,6]. Reading the VMS causes a drop in speed when approaching it. However, investing attention and time into reading and understanding the message is itself a distraction and therefore a risk. Furthermore, if we add the task of reading and understanding information to a complex task such as driving a car, we reduce the effectiveness of both tasks. There are solutions to reduce the attention required by simplifying information using pictograms or one-word messages. The latter are more effective for understanding messages than even pictograms, because understanding does not depend on prior knowledge of pictograms. There are conventions, such as the Vienna Convention, but each country is free to modify its signs, which makes it very difficult to recognize them quickly. There are also a few solutions such as READit VMS [7]

which, thanks to client–server architecture and user geolocation, locate the contents of the dashboard or display the images on the interior display of the vehicle. These applications require constant geolocation and an internet connection to check the nearest VMS and may experience latency issues. They are also limited to VMS registered in the system. Because of these dependencies, they are not self-propelled systems that allow the vehicle to be independent wherever it moves. The most similar ADAS are traffic-light recognition systems that, using sophisticated machine learning and computer vision techniques, display signals to the driver on a display on the dashboard. For this reason, one solution would be to use a machine learning-based approach that works with any VMS, regardless of whether it is collected in a database or not. In [8], an ADAS system that detects these signals and locates them automatically regardless of connectivity or the installation of new panels is proposed. The assistant consists of a processing pipeline that first recognizes the VMS in the scene, then processes the panel to extract the text by OCR, and finally announces it.

In [8] the first task was to create a dataset of images with the location of the VMS to properly train the supervised machine learning algorithms on which it is based, since there was no previous dataset of these characteristics; there were only datasets with their locations but no images [9,10]. Supervised learning consists of training the models using previously catalogued examples. Image labeling for training machine learning algorithms is an eminently manual task, so it is time consuming. At the time of writing this paper, there is, in the authors' opinion, no free dataset with annotated images that can be used to train Machine Learning models for VMS recognition on images captured by a vehicle camera such as those discussed above. This publication presents a free-to-use Spanish labeled VMS dataset, using a methodology that allows to increase the number of instances with as little manual work as possible.

## 2. Data Description

The dataset is constituted of 1216 instances (888 VMS and 328 negative examples) in 1152 color images in jpg format (Figure 1 shows a subset), each of which is complemented by an XML (Extensible Markup Language) file, according to the PASCAL VOC (Visual Object Classes) format, with the annotations of the location of the VMSs within the image, and other information. The structure of this file is as follows:

```
<annotation>
<folder></folder>
<filename></filename>
<path></path>
<source>
<database></database>
</source>
<size>
<width></width>
<height></height>
<depth></depth>
</size>

<object>
<name></name>
<pose></pose>
<truncated></truncated>
<difficult></difficult>
<bndbox>
<xmin></xmin>
<ymin></ymin>
<xmax></xmax>
<ymax></ymax>
```

```
    </bndbox>
  </object>
</annotation>
```

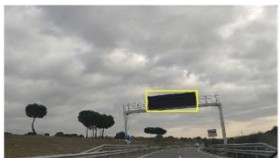 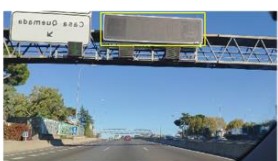 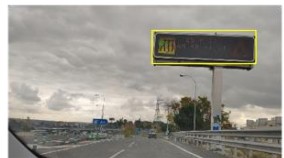 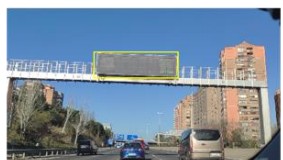

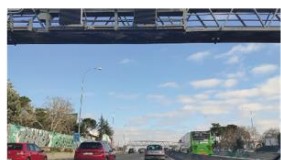 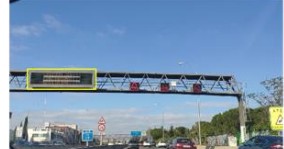 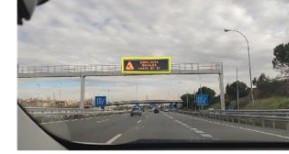 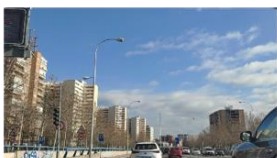

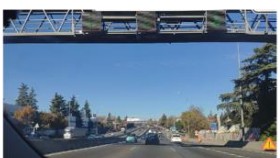 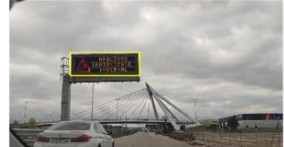 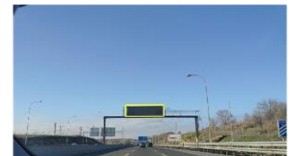 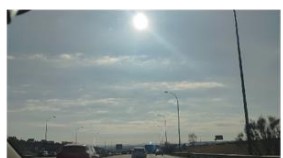

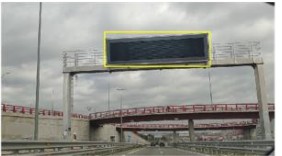 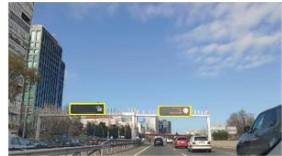 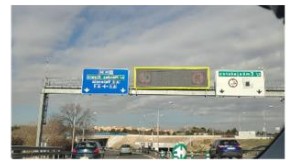 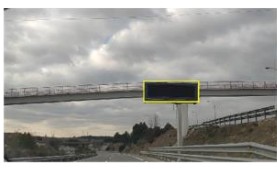

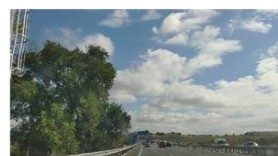 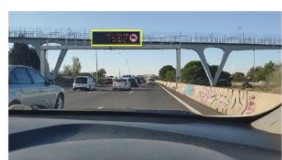 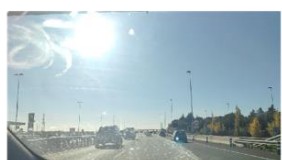 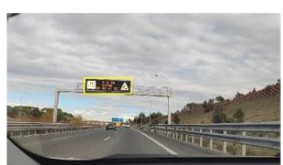

**Figure 1.** Dataset examples.

Although PASCAL VOC offers many fields, the ones used are the following:

- **folder:** Indicates the folder where the images are located.
- **filename:** Name and extension of the image file to which the annotation file refers.
- **path:** Absolute path of the image file after annotation.
- **size:** Size in pixels and number of channels. Color images have three channels while black and white images have one channel.
- **object:** It contains the data related to the object located in the image. This label and its contents are repeated for every single object located.
  - **name:** Object class name.
  - **bndbox:**
    - **xmin:** x-coordinate top left corner.
    - **ymin:** y-coordinate top left corner.
    - **xmax:** x-coordinate bottom right corner.
    - **ymax:** y-coordinate bottom right corner.

Some of the images have been created synthetically using data-augmentation techniques. Within the dataset folder, the real images (576) and the synthesized images (576) are separated in different subfolders.

The folder structure of the dataset is as follows:

- vms_dataset/
  - ○ data.csv
  - ○ real_images/
    - ▪ imgs/
    - ▪ annotations/

    [leftmargin=21pt,labelsep=12pt]
  - ○ data-augmentation/
    - ▪ imgs/
    - ▪ annotations/

In which:

- **data.csv:** Each row contains the following information separated by commas (,): image_name, x_min, y_min, x_max, y_max, class_name, lat, long, folder, text.
- **real_images:** Images extracted directly from the videos.
- **data-augmentation:** Images created using data-augmentation
- **imgs:** Image files in .jpg format.
- **annotations:** Annotation files in .xml format.

### 3. Methodology

The strategy is to create a processing pipeline that involves the least amount of manual work. For this reason, a process has been designed in which a minimum dataset will be obtained first, to create a basic model to iteratively process and label the images (Figure 2). Although the first part of the process will be completely manual, subsequent parts will consist of minor adjustments on images extracted from videos, which is less time consuming than labeling from scratch.

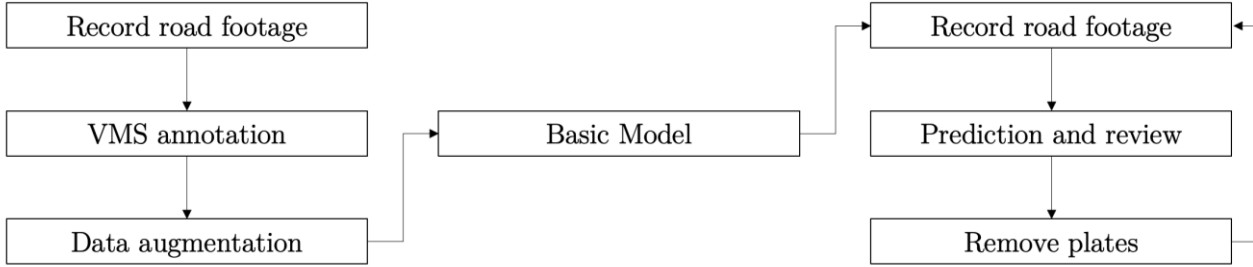

**Figure 2.** Methodology followed to obtain the dataset.

In addition, to increase the number of total instances, the data-augmentation technique will be used. This is a very extended method and consists of applying modifications to the image (rotations, cropping, translations, etc.) to create apparently new instances. In this case, since the VMS will always be at the top position of the image, we have chosen to flip the image on the $y$-axis. In this way, the panels on one side will be placed on the opposite side, generating a new instance.

**Record road footage.** The first task is to collect some videos of road routes around Madrid, Spain. These have been taken from inside a vehicle using the camera from a Xiaomi Mi 8 cellphone, along several routes through the main highways of Madrid in sunny and cloudy conditions during central daylight hours. The exact location of each VMS is indicated in the file data.csv.

**VMS localization annotation.** Once the first videos are obtained, several frames are extracted using a Python script (experimentally, every 50 frames the VMS image is different

enough to be considered as a new instance) and then subjected to a manual review process, eliminating those that do not have enough quality. Then, for each image, the location of the VMS is manually annotated using the software [11], which generates an XML file in PASCAL format.

**Data augmentation.** Once the images are annotated, using a Python script and the OpenCV library [12], a series of synthetic examples are created that appear to be entirely new instances. This is a popular technique to increase the number of examples with minimal effort. In this case, since the VMS is always at a certain height from the ground, the data augmentation used is a horizontal flip (around the y-axis).

**Basic model.** Once the dataset has enough images to create the basic model, it is trained using [13]. The selected model is a RetinaNET [14], a one-step CNN (Convolutional Neural Network) which has already proven its effectiveness [15] for this task in [8], with a pretrained resnet backbone. The mean average precision (mAP) has been established as the parameter to be optimized. Tables 1 and 2 show the hardware used, the training parameters and the result obtained, respectively. Figure 3 graphically illustrates the training results.

**Table 1.** Hardware used for training.

| Component | Name |
| --- | --- |
| Processor | Intel i7 9800K 3.6 GHz |
| RAM | 32 GBs |
| Graphics card | Nvidia RTX 2080 Ti |
| Hard disk | 1 Tb SSD M2 |
| OS | Ubuntu 18.04.4 LTS |

**Table 2.** Training params and result.

| Metric | Value |
| --- | --- |
| Intersection over Union (IoU) | 0.5 |
| Learning rate | $10^{-5}$ |
| Test/train split | 80–20% |
| Freeze backbone | True |
| Steps | 58 |
| Batch size | 8 |
| Total epochs | 23 |
| Final mAP | 0.976 |

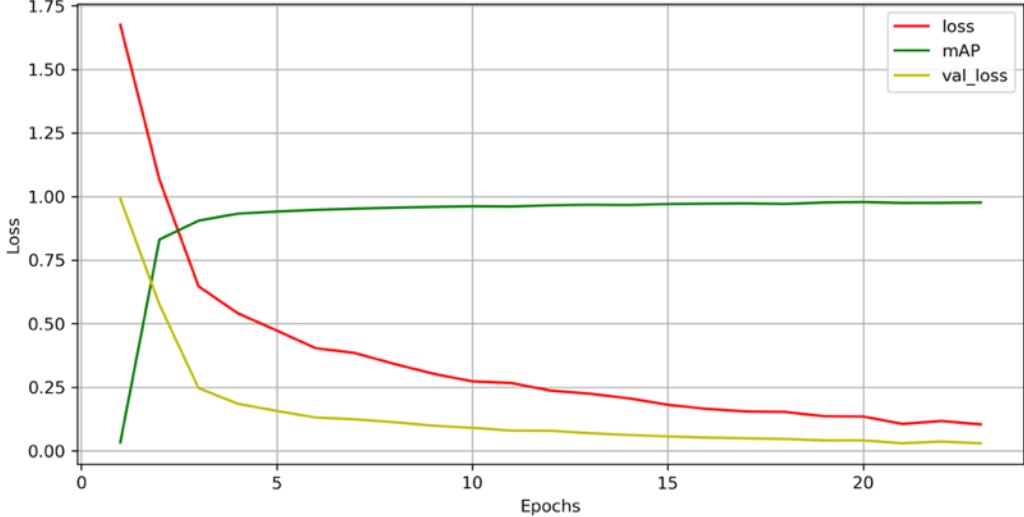

**Figure 3.** Training results.

**Adding more instances.** Thanks to the model, the task of increasing the dataset is easier and faster. To add new instances the process is: 1. Record new videos, extract frames and remove those with poor quality, 2. Predict the location of the VMSs using the model, 3. Review and confirm the images and predictions, and 4. Use the data-augmentation script to increase the size of the dataset.

The last step after completing the dataset is the anonymization of the images to comply with the European legislation [16] concerning data protection. Car license plates must be removed to make it impossible to read, as this information is considered personal data. Using a simple editing program this is achieved as not all images clearly show a legible license plate.

## 4. User Notes and Discussion

This work seeks to provide a dataset of VMS, using the least manual work, to train machine learning models to detect them on images and even to transcribe them to text. Since no such dataset exists, this work starts a new research line for the detection of these systems. The aim of this work is to provide researchers with a basic and very necessary ingredient for building Machine Learning models for VMS recognition in road traffic images: the set of annotated examples needed to train the algorithms. This dataset can therefore be used in Deep Learning and Natural Language Processing tasks for the extraction of the text from the signal messages.

As a next step for this dataset, it would be interesting to increase the number of instances in other conditions such as rain and low light.

**Author Contributions:** E.P. has been responsible of coordinating and supervising the machine learning algorithms. G.D.-L.-H. was responsible for labeling the dataset and performing the experiments for data augmentation and machine learning. J.S.-S. was responsible for vehicle sensorization, video recording and image capture. J.F.-A. was responsible for drafting. All authors have read and agreed to the published version of the manuscript.

**Funding:** This publication is part of the I+D+i projects with reference PID2019-104793RB-C32, PIDC2021-121517-C33, funded by MCIN/AEI/10.13039/501100011033/, S2018/EMT-4362/"SEGVAUTO4.0-CM" funded by Regional Government of Madrid and "ESF and ERDF A way of making Europe".

**Institutional Review Board Statement:** Not applicable.

**Informed Consent Statement:** Not applicable.

**Data Availability Statement:** The data presented in this study are openly available in https://doi.org/10.5281/zenodo.5904211 (accessed on 30 March 2022) with doi: 10.5281/zenodo.5904211.

**Conflicts of Interest:** The authors declare no conflict of interest.

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
