# Peer review of "Dataset: Variable Message Signal Annotated Images for Object Detection"

_data_

Round 1
Reviewer 1 Report
VMS dataset introduced in this paper would be helpful for object detection task for autonomous driving.
Minor things to be improved:
- References need to be cited in the paper.
- There are incomplete sentences, i.e., at line 34 and 155.
Author Response
Dear Reviewer,
thank you very much for your work reviewing our work and for the comments and suggestions you have made for the improvement of the paper.
First of all we would like to comment that there was a failure in the automatic conversion from Word to PDF made by the journal's platform for document uploading. The PDF resulting from the conversion had formatting errors that are not in the original document that we uploaded to the platform. Among these errors we have observed that the bibliographical references have disappeared in the text and that some sentences are incomplete (which does not happen in the original document). We have reported this problem to the editor and they are going to investigate what has happened.
We have uploaded a revised version in an older Word format in case that was the problem, but if in the revised PDF document you notice that the references are still missing and some sentences are incomplete, please contact the editor so that he can send you a correct PDF version.
Many of the comments you make in your review are related to this problem I just mentioned above and I hope they do not occur in this new upload of the document to the platform.

Reviewer 2 Report
Dear authors,
you have collected and annotated a new variable message signal for object detection. 756 real images have been collected. The reviewer has some suggestions and concerns:
1, A dataset with 756 real images is only a moderate dataset, which does not make the dataset a significant contribution. A dataset with at least thousands of images could be considered.
2, Since this is a dataset/benchmark paper, more object detection models should be evaluated like CenterNet, CornerNet, DETR, Deformable DETR, FCOS, Swin Transformer, PVT, YOLOs, HRNet, EfficientDet, etc.
3, Please discuss existing datasets and discuss the relations and differences between your dataset and existing ones. It would be nice to have a table to compare different datasets.
4, The diversity of the dataset should be described, in terms of collecting locations, weathers, day/night, and other conditions.
5, Please present computation complexity analysis results, which help gather a better understanding of the task, the dataset, and the model.
6, What data augmentation methods have been used? This should be discussed.
7, The effects of different data augmentation methods used in your model should be evaluated and analyzed.
8, Where and how would you make the dataset publicly available and release the dataset? Would you provide a benchmark with leaderboards with evaluation tools? Please discuss this.
For these reasons, we suggest that a major revision is required.
Sincerely,
Author Response
Dear Reviewer,
thank you very much for your work reviewing our work and for the comments and suggestions you have made for the improvement of the paper.
First of all we would like to comment that there was a failure in the automatic conversion from Word to PDF made by the journal's platform for document uploading. The PDF resulting from the conversion had formatting errors that are not in the original document that we uploaded to the platform. Among these errors we have observed that the bibliographical references have disappeared in the text and that some sentences are incomplete (which does not happen in the original document). We have reported this problem to the editor and they are going to investigate what has happened.
We have uploaded a revised version in an older Word format in case that was the problem, but if in the revised PDF document you notice that the references are still missing and some sentences are incomplete, please contact the editor so that he can send you a correct PDF version.
Below are the changes we have made and the responses to your comments:
1. A dataset with 756 real images is only a moderate dataset, which does not make the dataset a significant contribution. A dataset with at least thousands of images could be considered.
We agree that the more images, the better. But since this is a dataset for object detection with a single class, we think that 756 samples is a good enough number for training a competitive model.
According to Saleh Shahinfar, et al. (2020) in “How many images do I need?” paper, training a model shows an inflection point of around 150–500 images per class.
2, Since this is a dataset/benchmark paper, more object detection models should be evaluated like CenterNet, CornerNet, DETR, Deformable DETR, FCOS, Swin Transformer, PVT, YOLOs, HRNet, EfficientDet, etc.
We are not sure we understand this comment. Our paper presents a dataset but does not attempt to perform a benchmark with it, as that is totally beyond the scope of this journal. As noted in the description of this journal (MDPI Data), "Data articles are short descriptions of data and should contain no conclusions or interpretive insights". The suggestion to perform the experiments you propose is very interesting, but we think it does not suit the scope of this journal.
3, Please discuss existing datasets and discuss the relations and differences between your dataset and existing ones. It would be nice to have a table to compare different datasets.
During the previous steps of the dataset preparation, we carried out an exhaustive search for similar datasets of VMs and did not find any in the scientific literature or in public dataset repositories. This aspect of the uniqueness of our dataset has been added to the paper in chapter 4 "user notes".
4, The diversity of the dataset should be described, in terms of collecting locations, weathers, day/night, and other conditions.
The different locations and weather conditions have been added to the "Record road footage" section (lines 138-).
5, Please present computation complexity analysis results, which help gather a better understanding of the task, the dataset, and the model.
As already stated in the response to comment 2, this article presents the dataset but has not carried out an exhaustive experimentation on the data since it is not the objective of the publication, nor of the subject matter of the journal. For the same reason, no systematic information has been collected on the computational complexity of the data processing beyond the information relevant to the data capture.
6, What data augmentation methods have been used? This should be discussed.
7, The effects of different data augmentation methods used in your model should be evaluated and analyzed.
Data aumentation techniques are discused in chapter "Data-aumentation" (line 151).
In addition to the "Data-augmentation" section, this aspect is also discussed in the methodology section (line 130), and in the "Adding more instances" section of the paper.
8, Where and how would you make the dataset publicly available and release the dataset? Would you provide a benchmark with leaderboards with evaluation tools? Please discuss this.
The dataset is publicly available in the Zenodo repository (https://zenodo.org/record/5904211). Although this availability is indicated in the paper in the "Data Availability Statement" section (line 213), we have reinforced the visibility of the link also at the beginning of the document, just after the abstract.

Reviewer 3 Report
This paper tackles the need for annotated figure-based data sets to improve research related to computer vision. Especially the part related to object recognition and classification based on machine learning. This is a very actual topic making the paper's strong part and denoting its suitability to the chosen journal. However, this is a journal paper, and the paper structure should be organized to reflect all the needed discussion items. Namely, a clear scientific contribution description should be included, related work analysis denoting open issues, and a description of how your proposal addresses these open questions. An annotated data set is a scientific contribution, especially in a field where such data sets are needed. But, everything has to be elaborated accordingly.
Areas for improvement include:
- Always cite relevant literature when stating crucial claims regarding your paper topic;
- Check the paper writing as on some places there are extra blanks, and some sentences are not complete;
- Add a paragraph into the Introduction section explaining the paper contribution compared to existing annotated data sets;
- A journal paper should always have a strong state of the art or related work section emphasizing the improvement proposed in the paper;
- Elaborate on the recording framerate and how you decided to use every 50th frame in the extraction process;
- Add a conclusion section wrapping up the contribution of the paper, pros and cons, and future work on this topic;
- Add a discussion section commenting on your paper's achieved results and limitations.
There are additional comments in the attached PDF.

Author Response
Dear Reviewer,
thank you very much for your work reviewing our work and for the comments and suggestions you have made for the improvement of the paper.
First of all we would like to comment that there was a failure in the automatic conversion from Word to PDF made by the journal's platform for document uploading. The PDF resulting from the conversion had formatting errors that are not in the original document that we uploaded to the platform. Among these errors we have observed that the bibliographical references have disappeared in the text and that some sentences are incomplete (which does not happen in the original document). We have reported this problem to the editor and they are going to investigate what has happened.
We have uploaded a revised version in an older Word format in case that was the problem, but if in the revised PDF document you notice that the references are still missing and some sentences are incomplete, please contact the editor so that he can send you a correct PDF version.
Many of the comments you make in your review are related to this problem I just mentioned above and I hope they do not occur in this new upload of the document to the platform.
The decision to take one frame every 50 was decided experimentally by using different values. 50 is the number at which there was an appreciable change in the image with respect to the previous one but without losing context information.We have added this explanation to the document to justify the decision.
We have also strengthened the introduction and conclusions (section "user notes") as you suggested in your comments.

Round 2
Reviewer 2 Report
Dear authors,
thank you for your revision and the responses. The revision does solve some concerns. However, as a dataset paper, it is important to clearly present what new challenges are introduced. You have claimed that in the abstract "The purpose of this dataset is to help in road computer vision research". For this purpose, it is important to use different models for testing and show their detection performances. This can help understand how your dataset has helped computer vision research. This can also help understand the complexity and the novel problems introduced by your dataset. Please consider including some visualization of detection results based on different computer vision models. From the computer vision application perspective, you could also consider showing how well the model trained on your dataset generalizes to different conditions, e.g., different weather and day/night conditions or images collected in other cities/countries.
Sincerely,
Author Response
Thank you very much for your comments and suggestions to improve the paper.
In response to your comments, the following changes have been made to the document: The introduction has been substantially modified, with greater emphasis on the novelty and necessity of this dataset (lines 31 to 56 and 60 to 65).
Although this article only presents the dataset, the experimentation and model building part has been developed in another article already published by MDPI (reference 8). This reference has been emphasized in the paper, especially in the introduction for those readers who want to go beyond the dataset and consult the results of different ML models (lines 52 to 67).
Reviewer 3 Report
The authors have improved the paper. However, three crucial comments were not covered in the paper augmentation. I am aware that the journal Data accepts data set descriptions, but a clear description about the novelty of the proposed data set compared to the state of the art should be given.
Areas for improvement include:
- Add a paragraph into the Introduction section explaining the paper contribution compared to existing annotated data sets or emphasizing the need for a new data set;
- A journal paper should always have a strong state of the art or related work section emphasizing the improvement proposed in the paper;
- Add a discussion section commenting on your paper's achieved results and limitations.
There are additional small comments in the attached PDF.

Author Response
Thank you very much for your comments and suggestions to improve the paper.
In response to your feedback, the following changes have been made to the paper: The introduction has been substantially modified, emphasizing the novelty and necessity of this dataset (lines 31 to 56 and 60 to 65).
A discussion paragraph has been added at the end of the document, commenting on the results and contributions of the dataset proposed in this paper (lines 202 to 210).
Round 3
Reviewer 2 Report
Dear authors,
This work seeks to provide a dataset for the community and to show how to make use of the dataset for object detection. As a journal paper and a dataset paper, it is important to provide comprehensive results to support your claim that the dataset can advance the field. Only using a basic model is clearly not enough to support your claim. The reviewer believes that some more state-of-the-art object detection models should be used, tested, and benchmarked based on your dataset. If this is not shown in the paper, then you obviously do not need a journal paper. Only a website or a GitHub repository can already show the necessary information about the dataset.
Sincerely,
Reviewer 3 Report
The paper can now be accepted and it present and well-rounded scientific presentation of a needed dataset.